# The Mechanistic Action of Biosynthesised Silver Nanoparticles and Its Application in Aquaculture and Livestock Industries

**DOI:** 10.3390/ani11072097

**Published:** 2021-07-14

**Authors:** Catrenar De Silva, Norazah Mohammad Nawawi, Murni Marlina Abd Karim, Shafinaz Abd Gani, Mas Jaffri Masarudin, Baskaran Gunasekaran, Siti Aqlima Ahmad

**Affiliations:** 1Department of Biochemistry, Faculty of Biotechnology and Biomolecular Sciences, Universiti Putra Malaysia UPM, Serdang 43400, Selangor, Malaysia; catrenar94@gmail.com (C.D.S.); shafinaz_abgani@upm.edu.my (S.A.G.); 2Institute of Bio-IT Selangor, Universiti Selangor, Jalan Zirkon A7/A, Seksyen 7, Shah Alam 40000, Selangor, Malaysia; norazah@unisel.edu.my; 3Centre for Foundation and General Studies, Universiti Selangor, Jalan Timur Tambahan, Bestari Jaya 45600, Selangor, Malaysia; 4Department of Aquaculture, Faculty of Agriculture, Universiti Putra Malaysia UPM, Serdang 43400, Selangor, Malaysia; murnimarlina@upm.edu.my; 5Laboratory of Sustainable Aquaculture and Aquatic Sciences, Port Dickson 71050, Negeri Sembilan, Malaysia; 6Department of Cell and Molecular Biology, Faculty of Biotechnology and Biomolecular Sciences, Universiti Putra Malaysia UPM, Serdang 43400, Selangor, Malaysia; masjaffri@upm.edu.my; 7Department of Biotechnology, Faculty of Applied Sciences, UCSI University, Taman Connaught, Kuala Lumpur 56000, Malaysia; baskaran@ucsiuniversity.edu.my; 8Laboratory of Bioresource Management, Institute of Tropical Forestry and Forest Products (INTROP), Universiti Putra Malaysia UPM, Serdang 43400, Selangor, Malaysia

**Keywords:** biotechnology, nanotechnology, toxicity, aquaculture, livestock, poultry

## Abstract

**Simple Summary:**

Silver nanoparticles (AgNPs) have been employed in various fields of studies due to their impeccable ability as an antibacterial, antifungal and antiviral agent. AgNPs are generally synthesised via three methods, which are the chemical, physical and biological methods, with the biological method being the preferred method recently due to the capability of synthesizing nanoparticles of various shapes and sizes to enhance antimicrobial activity and reduced environmental effects. In recent years, AgNPs have been employed in the aquaculture, livestock and poultry industries to combat pathogens. This review is centred on the cytotoxic mechanistic action of AgNPs, which contributes to its application against pathogens in the aquaculture, livestock and poultry industries.

**Abstract:**

Nanotechnology is a rapidly developing field due to the emergence of various resistant pathogens and the failure of commercial methods of treatment. AgNPs have emerged as one of the best nanotechnology metal nanoparticles due to their large surface-to-volume ratio and success and efficiency in combating various pathogens over the years, with the biological method of synthesis being the most effective and environmentally friendly method. The primary mode of action of AgNPs against pathogens are via their cytotoxicity, which is influenced by the size and shape of the nanoparticles. The cytotoxicity of the AgNPs gives rise to various theorized mechanisms of action of AgNPs against pathogens such as activation of reactive oxygen species, attachment to cellular membranes, intracellular damage and inducing the viable but non-culturable state (VBNC) of pathogens. This review will be centred on the various theorized mechanisms of actions and its application in the aquaculture, livestock and poultry industries. The application of AgNPs in aquaculture is focused around water treatment, disease control and aquatic nutrition, and in the livestock application it is focused on livestock and poultry.

## 1. Introduction

Nanotechnology has gained attention since the 1980s and has developed rapidly in the years to follow, which has been proven impeccable in combating various pathogens and human diseases in various fields such as medicine, pharmaceuticals and the food industry [1,2]. The most common application of nanotechnology is the generation of nanoparticles such as organic, non-organic, polymer-based, non-polymeric and metallic nanoparticles. In recent years, metallic nanoparticles have gained attention in the field of nanotechnology due to their ability to offer a range of surface modifications, which contributes to the nanoparticles being more stable, biocompatible and confers specific functionalities needed for applications in various fields. Researchers have successfully synthesised metallic nanoparticles via physical, chemical and biological synthesis methods, of which biological synthesis is the most preferable method due to its eco-friendly and low-cost approach compared to physical and chemical means that generate nanoparticles at a high-cost rate from the use of various types of machinery and chemicals, thus resulting in environmental issues. The biological synthesis method generates nanoparticles of various shapes and sizes [2,3], and the primary mode of action of nanoparticles that gives them their multidisciplinary functions is their small size (1–100 nm) and their shape (spherical, triangular or rod).

Silver nanoparticles (AgNPs) are one of the many nanoparticles that have been employed in various fields. AgNPs are the nano-sized version of bulk silver that possesses the same antibacterial activity as their large counterpart, but with better efficiency and effectiveness due to their small size and shape, which contribute to their large surface-to-volume ratio, enhancing their antibacterial potential. Over the years, AgNPs have gained attention as one of the most successful nanoparticles that have been applied in many fields, demonstrating the highest success as antibacterial agents. However, the application of AgNPs in aquaculture and livestock is relatively new; there are in vitro results, yet in vivo results are lacking [1]. The application of AgNPs in aquaculture and livestock in crucial to combat pathogens that pose a threat to the industry, resulting is loss of product and leading to economic losses.

Due to this major issue, study into silver as an antibacterial agent has been accelerating over the years, specifically research on AgNPs. AgNPs are promising antibacterial agents, are one of a kind and can significantly change their physical, chemical and biological properties due to their surface-to-volume ratio [4]. Among various methods of synthesis, the biological means of synthesis is by far the simplest; it is non-toxic, dependable, rapid and produces nanoparticles of well-defined size and shape under optimum conditions [5]. Silver nanoparticles trigger cell death via a primary action of cytotoxicity. The action of silver nanoparticles is complex and centred around their size and shape. This review will be centred on the mechanistic action of AgNPs and its application in the aquaculture and livestock industries.

## 2. Properties of AgNPs Based on Size and Shape

The toxicity of AgNPs depends primarily on their size and shape; other factors such as surface charge, functionalisation and core structure are also crucial for the biological action of the nanoparticles [6,7]. These factors play a crucial part in the mechanism of action towards bacterial cells in terms of cellular uptake, cellular activation and intracellular distribution [6,7]. Researchers have argued over how size and shape play a key role in AgNP toxicity towards bacterial cells, and it was concluded that small nanoparticles with sharp edges or facets are able to better penetrate and kill bacterial cells [8,9].

### 2.1. Size of AgNPs (Size Dependent)

The main characteristics studied were particle size and antibacterial activity. It was found that the antimicrobial activity of nanosilver was closely related to size. In their study, Morones et al. [8] concluded that AgNPs, mainly of smaller size ranging between 1 and 10 nm, attached to the cell membrane of target bacteria and interfered with cell functions, primarily penetration and bacterial respiration. The nanoparticles were able to penetrate the bacterial cell and cause severe damage to intracellular mechanisms, leading to cell death by interacting with bacterial DNA. Morones et al. [8] also summarized that AgNPs of smaller size released silver ions, which has an added advantage to antibacterial effects of nanosilver. The study by Feng et al. [10] supported that the interaction of silver ions with thiol proteins interrupted the enzymatic activity of bacterial cells. Meanwhile, Sotiriou et al. [11] argued that nanosilver of smaller size releases silver ions faster as compared to larger nanoparticles, which leads to higher toxicity due to more effective silver ions. El-Nour et al. [12] also reported that the smaller the size of silver nuclei, the higher the antibacterial potential displayed. Table 1 shows the strains of pathogenic bacteria on which antibacterial testing was done and the specific nanoparticle size responsible in killing the bacterial strains.

### 2.2. Shape of AgNPs (Shape Dependent)

In recent years, studies have focused on nanoparticle size and how size is the primary means for nanoparticles to penetrate the bacterial cell and kill target bacteria. However, more researchers have ventured into studies related to the shape of the nanoparticles and how shape and size are correlated in the mechanism of antibacterial action of nanoparticles. The antibacterial activity of nanosilver with different shapes has been discussed by Pal et al. [20] and Dong et al. [9]. The most common shape of nanoparticles is circular. Since venturing into nanoparticles-related research, researchers have discussed how circular-shaped nanoparticles can easily penetrate bacterial cells by passing through protein channels on the plasma membrane. Huang et al. [21] were able to optimize and synthesize AgNPs of circular shape, which were effective against the plant pathogen *Bipolaris maydis*. However, recent research suggests triangular AgNPs exhibit a higher inhibition activity against bacteria compared to circular AgNPs due to the presence of a basal plane, which gives the triangular AgNPs a stronger antibacterial activity against bacteria at a high atom density [20]. Dong et al. [9] added to this by synthesising triangular nanoparticles with sharper vertexes and edges and tested the antibacterial potential against bacteria with success. Thus, they hypothesised that a geometrically triangular Nano prism, having a very sharp apex and sharp edges, would better penetrate and damage bacterial cells [9]. Figure 1 shows various shapes of nanoparticles with sharp vertexes. AgNPs with different shapes would also have diverse effects on the bacterial cell. Besides spherical and triangular shapes, nanoparticles can be synthesised as cubes, platelets, ovals, hexagons and rods.

Numerous methods have been developed to synthesize nanosilver in various shapes. AgNPs can be structurally synthesized in the form of cubes, rods, platelets, pyramids and bipyramids [24,25]. The difference in shape results is the difference in efficacy of the AgNPs. As reported by Dong et al. [9], shapes with edges, specifically sharp edges, are able to penetrate the bacterial cell easier compared to spheres, which have no edges. Hong et al. [15] reported that AgNPs in the form of cubes were most effective against *Escherichia coli* as compared to spheres and wires. Pal et al. [20] reported comparable results for rod, circular and triangular AgNPs against *Escherichia coli*. However, there are reports of no antibacterial activity from AgNPs in the form of cubes, spheres and triangles against *Staphylococcus aureus* [26]. The published results obtained from research into shape-dependent nanoparticle antibacterial activity have been inconsistent; therefore, further research is required to fill the existing gaps [25].

Nanoparticle size and shape work in correlation; the smaller the nanoparticle and the sharper the edges, or the more edges present on the nanoparticle, the easier the penetration into the bacterial cell. The combination of these two factors makes AgNPs an effective antibacterial agent [9].

## 3. Mechanism of Action of AgNPs

The mechanism of action of AgNPs on bacterial cells has been yet to be fully understood. Researchers have theorised on possible mechanisms that may be related to changes of the morphology and structure of the bacterial cell caused by the size of nanoparticles in relation to the large surface-to-volume ratio and their shape [27]. These physiochemical factors play a crucial role in the antibacterial action of AgNPs [28,29]. The small size of the nanoparticles provides better interaction with the bacterial cell and ease in penetrating the bacterial cell [27]. However, for the specific size and shape production of AgNPs, factors including temperature and pH are important [30]. Application of heat over a time period can increase the size of nanoparticles as well as change their shape, as observed by Mokhena et al. [31] after heating AgNPs at 90 °C for 3 h, which resulted in the increase in the size of the nanoparticles from 28 nm to 30 nm while the shape of the nanoparticles changed from spherical to an irregular shape. Meanwhile, further heating for 48 h resulted in the formation of a mixture of rod-like nanoparticles with sizes between 76 nm and 121 nm as well as spherical nanoparticles of sizes between 28 nm and 56 nm. Huang et al. [32] were able to synthesise AgNPs of sizes ranging from 8 nm to 24 nm via heating up to 80 °C for 15 min.

Another theory argued by researchers is that the antimicrobial action of AgNPs is similar to silver ions due to the former being oxidised into the latter [8,33,34,35]. Xiu et al. [35] reported that the antimicrobial action of AgNPs is oxygen-dependent and ineffective in the absence of oxygen, primarily being that the conversion of AgNPs to silver ions occurs in the presence of oxygen, and the defined molecular toxicants are silver ions. They then tested the antimicrobial action of nanosilver by synthesizing it in anaerobic conditions, and they observed the lack of toxicity of the AgNPs due to the reduction in the formation of silver ions.

### 3.1. Production of Reactive Oxygen Species (ROS)

Cell death is induced by ROS. ROS causes cell death in two separate ways: apoptosis or necrosis. The apoptotic course is activated by the ROS through caspases, which are the killers of apoptosis [36]. The activation of caspases is determined as the point of no return in apoptosis [37]. The function of caspases in the apoptosis pathway is to cleave DNA of the bacteria, which is an onset of apoptosis.

Silver ions produced from the oxidation of AgNPs catalyse the production of ROS [38,39]. Metal nanoparticles have been reported to induce a significant rise in ROS in cells inducing toxicity related to oxidative stress [40,41,42]. Oxidative stress can be triggered by disruption on the respiratory chain of the targeted bacteria or by the silver nanoparticles [43]. ROS possibly result from the interaction of silver ions with the thiol enzyme group during the inhibition of the respiratory chain via respiratory enzymes [44]. As a major factor affecting oxidative stress, ROS could cause damage to bacterial cell macromolecules such as protein synthesis and alteration, inhibition of enzymes, lipid synthesis as well as oxidation and damage of DNA and RNA of the bacterial cell [45]. At high levels, ROS can cause cell death, whereas severe damage or mutation to the DNA of the bacterial cells can occur at low levels.

Mats et al. [46] reported that ROS may react directly with the DNA or protein of bacterial cells or lipids of the bacterial cell producing malondialdehyde, which is a marker for oxidative stress that can, in turn, react with bacterial DNA, protein or lipids and cause cell damage or death. An early study by Messner and Imlay [47] reported that oxidative stress triggered by high temperatures of *Escherichia coli* treated with nanosilver resulted in the formation of ROS via the autoxidation of NADH dehydrogenase II in the respiratory chain, leading to cellular damage. AgNPs can directly interact with essential enzymes, induce nitrogen reactive species production and induce programmed cell death [29,48]. The full mechanism requires extensive research to be conducted. Until then, researchers can only theorise on the possible mechanisms of action. Figure 2 illustrates a simplified diagram of the possible mechanism of action towards Gram-negative and Gram-positive bacterial cells.

### 3.2. Attachment of AgNPs to the Cell Membrane of Bacterial Cells

AgNPs have the ability to adhere to the cell wall and penetrate it, which in turn causes changes to the structure of the cell membrane, including permeability and leading to cell death [33]. The AgNPs attach to the cell membrane by means of electrostatic charges, where the positive surface charge of the nanoparticle electrostatically adheres to the negative charge of the cell membrane, thus facilitating nanoparticle membrane attachment [50]. Upon interaction, an evident morphological change is observed and can be characterised by cytoplasm shrinkage and cell membrane rupture [50]. Raffi et al. [51] reported that, via transmission electron microscope (TEM) analysis, complete cell membrane disruption of *Escherichia coli* was observed upon minutes of exposure to AgNPs. Pits were observed around the areas of AgNPs damage induced by the AgNPs during attachment with the bacterial cell membrane [33].

Another factor is that AgNPs can interact with sulphur-containing proteins present on the cell wall, causing cell wall damage [52]. This in turn affects the lipid bilayer and permeability of the cell membrane, which affect the bacterial cells’ ability to regulate transport through the membrane [53]. A study has been done on *E. coli* to examine this factor. Gram-negative bacteria have an outer membrane outside the peptidoglycan layer, which is absent in Gram-positive bacteria. The outer membrane serves as a selectively permeable layer that only allows the entry of selective components for cell growth, whereas the layer serves as a protective barrier for the bacterial cell to prevent the entry of harmful substances that may temper bacterial cell activity and lead to cell death. The outer membrane is made up of liposaccharide (LPS) molecules with a small portion of membrane proteins. The LPS layer serves as a selectively permeable membrane for Gram-negative bacterial cells [45]. The presence of AgNPs may increase the permeability of the membrane, hence causing a reduction in intake of sugar and protein by the bacterial cell that in turn results in bacterial growth inhibition [54]. Schreurs and Rosenberg [53], in an early study, reported that silver can reduce the uptake and release of phosphate ions in *E. coli*. The transport and release of potassium ions can be altered from the bacterial cell. An increase in membrane permeability can also cause the leakage of cellular contents such as proteins, ions, reducing sugars and, in some cases, ATP molecules [55,56,57]. Figure 3 displays the pathway of cell death caused by increased permeability of the cell membrane based on the literature of Rai et al. [27].

### 3.3. Damage to Intracellular Components of Bacterial Cells

Once the AgNPs adhere to the cell membrane of bacterial cells, they can penetrate into the bacterial membrane and enter the cell, which affects important cell functioning [58,59]. Smaller-sized nanoparticles affect the intracellular structures of bacterial cells at a faster rate than bigger nanoparticles due to their large surface-area-to-volume ratio [60]. Once penetrated by the bacteria, nanosilver can interact with cell components such as proteins, lipids, enzymes and DNA. The interaction between the silver nanoparticles and the cellular components can lead to severe damage to bacterial cells such as dysfunction and eventually bacterial cell death. Interactions between AgNPs and bacterial cell ribosomes cause the denaturation of ribosomes that can inhibit protein synthesis [8,61,62]. Figure 4 illustrates intracellular damage of the cell caused by AgNPs.

AgNPs can also directly attach to functional groups of proteins, resulting in deactivation. For example, silver ions have been shown to bind to thiol groups of proteins present in the membrane of bacterial cells, thus forming stable bonds that result in the deactivation of protein molecules, leading to inhibition of ion transport across the membrane and transmembrane ATP generation [62,64]. Lok et al. [34] observed that nanosilver in its pure form or in the form of silver ions can alter the 3D structure of protein molecules present in bacterial cells; it interferes with the disulfide bonds in protein molecules and blocks the active binding side on protein molecules, leading to defects in the function of the bacterial cells. AgNPs have also been reported by Bhattacharya and Mukherjee [65] to interact and block sugar metabolism, which affects the glycolytic pathway of the bacterial cell and cause bacterial cell death. This was done by the interaction of AgNPs with the enzyme phosphomannose isomerase, which mediates the isomerisation of mannose-6-phosphate into fructose-6-phosphate that plays a key role as an intermediate in the pathway.

Direct interaction of AgNPs with DNA of the bacterial cells can lead to devastating effects in terms of cell division [66,67]. Klueh et al. [64] revealed that silver ions can interact with the nucleoside of the nucleic acid, imbed between the purine and pyrimidine base pairs and cause disruption of the double helix structure of DNA by disrupting the hydrogen bonds between the base pair of the anti-parallel DNA strand, which may block the transcription of genes by the bacterial cells [8]. Silver nanoparticles may also cause the DNA helix to become more condensed, which results in the bacterial cells losing their replication capacity [10].

### 3.4. Inducing the Viable But Non-Culturable (VBNC) State

Silver ions can have a viable but non-culturable (VBNC) effect on cells. The phenomenon by which bacterial cells are alive but unable to grow in standard conditions is known as the viable but non-culturable (VNCB) state [68,69]. In this stage, bacterial cells are unable to carry out cell division due to the inhibition of key factors such as uptake and utilisation of important substrates when the AgNPs are attached and cause structural and morphological changes to the bacterial cell membrane. These bacterial cells are metabolically active, but they show a reduced rate of nutrient transport, respiration and synthesis of macromolecules [45]. In some cases, bacterial cells in this state can stay alive for a certain period; however, they eventually die due to lack of nutrients and cell dysfunction. Jung et al. [61] reported that silver ions were able to inhibit the growth of *Escherichia coli* and *Staphylococcus aureus* and eventually led to bacterial cell death after 2 hours of exposure to silver ions. Although the exact mechanism of action for AgNPs has not been identified, results by various researchers have shown possible mechanisms required thorough investigation, observation and results.

## 4. Application of AgNPs in Aquaculture, Livestock and Poultry Industries

The application of AgNPs in aquaculture and livestock and poultry industries plays a crucial role in contributing to an increase in economic value through the reduction in issues related to aquatic and livestock diseases. Application of AgNPs in aquaculture can increase the survival rates and yield of aquatic life in ponds through water treatment; in livestock breeding, AgNPs can improve animal immunity through the reduction in the use of antibiotics and increased production of poultry [1]. Figure 5 illustrates the summary of the application of AgNPs in aquaculture, livestock and poultry industries. 

### 4.1. Application of AgNPs in Aquaculture 

The aquaculture industry is one of the rising industries in recent years that has to keep up with the demand for seafood required for the global population. Although the industry is relatively new as compared to farming and fishing industries, it has gained much attention. The industry faces a relatively serious issue of fish and shellfish-related diseases caused by pathogenic bacteria and viruses, which have caused a lot of economic losses to counter this issue. Besides diseases, water quality is also an issue that is correlated with the occurrence of diseases in aquaculture products. Although methods are available for treating fish and shellfish-related diseases, the methods do not stop the issue at hand but rather offer a temporary solution. Moreover, resistance to commercially available antibiotics has also occurred. Thus, a reliable and effective solution is required to overcome this problem and increase the production of aquaculture products. Nanotechnology, specifically AgNPs, offers an excellent solution and has been proven effective in in vitro testing and small-scale in vivo applications to overcome three major concerns, which are water treatment, disease control and aquatic nutrients [70,71].

#### 4.1.1. AgNPs in Water Treatment

Due to the large-scale production of the aquaculture industry, a large land area and high water quantities are required to keep up with supply and demand. As a result, aquaculture imposes an environmental risk to areas surrounding the aquaculture farm regardless of freshwater, seawater, brackish ponds or seawater cages. Water source is a limited and valuable source, and its use in aquaculture puts a strain on water resources. The contamination of ponds with unconsumed food, excretory by-products, chemicals and antibiotics generates pollution of pond water resulting in bacterial, fungal and viral infections. The common solution is continuous water changes, which causes a reduction in the water source as hundreds of cubic meters of water are needed per day depending on pond size and pollution of the surrounding environment with the polluted wastewater [72,73]. Thus, nanotechnology can offer a more cost-effective and efficient solution.

AgNPs offer a means to solve water pollution issues via remediation and water treatment using a filter system. Pradeep [74] reviewed the use of nanoparticles in water treatment, which showed success in eliminating bacterial and viral pathogens from water sources. However, another study concluded that the use of AgNPs directly into ponds reduced fungal infection on rainbow trout (*Oncorhynchus mykiss*), yet also caused the reduction in chloride and potassium in blood plasma in juvenile rainbow trout, suggesting that AgNPs need to be applied in other means to only combat bacterial and viral infections without causing harm on aquaculture products [75]. An effective solution to this problem is the use of AgNPs in filter systems. In one study, AgNPs were combined with zeolite and incorporated into filter systems with varying zeolite and AgNPs concentrations (the control was zeolite alone in a semi-recirculating system. Water was treated with the filter systems, and fertilised ovules of rainbow trout were added, with results showing a 5% increase in survival rate of fingerlings compared to the control test [75]. This proves that AgNPs are excellent water treatment agents and are best used as a filter system. Figure 6 shows a simplified diagram based on the study done by Sarkheil et al. [76] of the application of AgNPs in a water filtration system with a silver absorbent layer against *Vibrio* sp. strain Persian1 infecting Pacific White Shrimp. Two filter systems were designed, one with a silver absorbent and one without a silver absorbent. The results indicate that the filter system with the silver absorbent showed a significantly higher bacterial removal as compared to the filter system without the silver absorbent after 2, 6 and 12 h, respectively. 

#### 4.1.2. AgNPs in Disease Control

Bacterial, fungal and viral infections are major problems in the aquaculture industry. The commercially available method is the use of antibiotics to treat the infections; however, excessive application over time results in the emergence of resistant strains, making antibiotic treatments unsuccessful [77]. The most commonly used antibiotic based on data from 25 countries is tetracycline. Bacterial pathogens are also resistant to a wide range of antibiotics. The most common resistant bacterial strains are *Aeromonas salmonicida*, *Photobacterium damselae*, *Yersinia ruckeri*, *Listeria* sp., *Vibrio* sp., *Pseudomonas* sp. and *Edwardsiella* sp., which affect both freshwater and saltwater fish as well as shellfish species. AgNPs have been selected as an alternative to combat pathogens and produce efficient and effective results.

The use of biologically synthesised AgNPs has shown positive results in in vitro studies. AgNPs synthesised from tea plants showed 70% inhibition of *Vibrio harveyi* in infected Indian white prawns (*Feneropenaeus indicus*) at a concentration of 10 μg/mL [78]. Sivaramasamy et al. [79] synthesised AgNPs using *Bacillus subtilis* as a nanofactory and tested its antibacterial potential on *Vibrio parahaemolyticus* and *Vibrio harveyi* on white leg shrimp, showing a 90% inhibition. In another study, AgNPs encapsulated with starch at 10 mg AgNPs concentration was used to treat *Ichthyophthirius multifiliis* and *Aphanomyces invadans* fungal infections in fish species, and a recovery rate of three days was demonstrated, indicating inhibition or death of the fungal pathogens [80,81]. AgNPs were also able to be formulated into a vaccine to treat the white spot syndrome, a viral infection in shrimps, with success [82]. In another study, researchers were able to increase the mortality rate of shrimp (*Litopenaeus vannamei*) against the white spot syndrome virus by 50% with the application of the patented Agrovit-4^®^ (Novosibirsk, Russia; patent 2427380), a PVP-coated spheroid AgNP, at a dose of 1000 ng. A minimal dose of 10 μg/g was shown to be effective in protecting the shrimp, and an increase of 16% in mortality rate was observed as compared to the positive control, which had no survival [83,84,85]. The application of Argovit^®^ on the other hand showed an increased mortality rate of 70–80% (various doses) against the white spot syndrome virus in shrimp [83].

#### 4.1.3. AgNPs in Aquatic Nutrition 

The application of AgNPs in aquatic nutrition is fairly new yet employs the same technique of using AgNPs as nanocarriers to increase absorption of nutrients in fish and shellfish via encapsulation, targeted delivery and controlled release, or as cargo to a nanocarrier depending on their necessity. As a nanocarrier, AgNPs offer better adsorption and delivery of nutraceuticals required by aquatic species for growth. Biologically synthesised AgNPs are the best choice of nanoparticles due to their biodegradable ability at the end of the delivery pathway [86]. Alishahi et al. [87] reported that the application of nanotechnology in fish food pellets incorporating vitamin C resulted in better absorption of the nutrient in rainbow trout compared to the control with regular fish food pellets. This paves a platform for the further application of AgNPs in aquatic nutrition as it offers a reduced cost in fish food due to the smaller feeding portions and higher nutrient impacts, thus also reducing food wastage and reducing toxic effects of AgNPs.

### 4.2. Application of AgNPs in Livestock and Poultry

The livestock and poultry industry are the largest industries in the food sector due to the global demand for meat. Like the agricultural and aquaculture industries, the livestock and poultry industry also face a threat, which is diseases that affect the breeding and growth of livestock and poultry [1]. Nanotechnology offers a possible solution to the problem, as commercially available methods such as antibiotics have failed to counter the problem due to the emergence of resistance in bacterial pathogens. Colloidal silver has been used since the early 1950s as an additive in animal feed and showed improved growth and reduction in infections, yet the use of colloidal silver was stopped and replaced with a cheaper alternative known as antibiotics [88]. In recent years, in vitro research has been done on the use of AgNPs to kill pathogenic bacterial cells and as a feed additive for antibacterial effects to promote animal growth.

#### 4.2.1. Application of Silver Nanoparticles in the Livestock Industry

The livestock industry plays a major role in contributing to economic growth; however, over the past decade the quantity and quality of livestock products have reduced due to the emergence of resistant strains of pathogenic microorganisms failing commercial methods in combating said pathogens. Foodborne pathogens *Escherichia coli*, *Staphylococcus aureus*, *Streptococcus agalactiae*, *Pseudomonas aeruginosa*, *Salmonella enterica* and *Klebsiella pneumoniae* are some species that cause diseases that result in increased mortality rates in livestock such as cows, goats, sheep and pigs [89]. AgNPs offer an effective and efficient solution to existing issues of resistance primarily due to the mechanism of action of AgNPs, which plays a crucial role in the treatment of pathogenic bacteria. The primary mechanism of action of AgNPs is their toxicity, via the release of silver ions, and activation of oxygen reactive species that trigger oxidative stress; both put stress on the bacterial components, such as protein synthesis and DNA formation, and result in deactivation or cellular death. 

Sondi and Salopek-Sondi [33] and Jung et al. [61] observed cellular death via inhibitory effects of AgNPs on *Escherichia coli*. Meanwhile, Li et al. [90] and Kim et al. [91] observed similar effects of AgNPs against *Staphylococcus aureus*. AgNPs have also been applied as an additive in animal feed, where Fondevila et al. [92] observed the reduction in coliforms in pig microbiota and an increase in pig growth rate when the AgNPs colloid was used as a supplement in the feed. The results indicated the death of harmful microorganisms (coliforms) and no effects on healthy gut bacteria of the pigs. They also reported that an increase in the dose of AgNPs from 0 to 40 mg/kg resulted in an increase in the growth rate of weaned pigs in a period of 28 to 56 days [92]. In another study done by Kalinska et al. [93], in vitro results of the application of AgNPs as a potential treatment/prevention against mastitis in dairy cows and goats, caused by pathogenic species such as *Escherichia coli*, *Staphylococcus aureus*, *Enterobacter cloacae*, *Enterococcus faecalis, Streptococcus agalactiae* and *Candida albicans*, showed strong antibacterial and antifungal activity in pure AgNPs and with a combination of silver and copper nanoparticle complex (AgCuNP). AgNPs showed a stronger antibacterial and antifungal activity as compared to the AgCuNP complex with pathogen viability of *p* < 0.01.

#### 4.2.2. Application of Silver Nanoparticles in the Poultry Industry

The poultry industry faces the tremendous loss of products primarily due to pathogenic infections brought on by pollution of water sources used in poultry farms as well as improper handling, which in turn causes a decrease in growth rate and increase in mortality. AgNPs have proven successful in combating pathogenic infections and reducing the mortality rate in poultry. AgNPs can be introduced to target organisms via various methods such as in feed and water source, by injecting target organisms and by tropical applications such as spraying or dipping in the AgNPs colloid for eggs. Figure 7 illustrates the basic methods of application of AgNPs on poultry.

Studies have shown that AgNPs can increase the weight of poultry by acting as a growth promoter as well as an effective antibacterial agent by targeting the cell membrane and causing DNA damage via direct and indirect routes against pathogens that infect poultry. Authors in [27,94,95] reported that groups of quail that consumed AgNPs in feed and water had an increase in body weight from 40 g to 98.9–102.2 g after 12 days of the study. AgNPs with a size range of 1 to 100 nm at lower concentrations (<70 ppm) have shown to be effective in combating pathogens and induce growth of poultry with minimal to no toxic effects as compared to higher concentrations of the same size of AgNPs [96].

Sawosz et al. [94] reported that the application of 50 ppm of colloidal AgNPs to chicken embryos caused no effects on the growth, development and mortality of chicken embryos after 48 h and after 20 days. Biochemical tests of blood serum of the chicken embryos showed no effects to the liver. However, increased mineral content was noticed, indicating that AgNPs influence bone mineralization. In another study, chicken eggs were treated with 50 ppm colloidal AgNPs. The results obtained indicated no pathogenic infection or spoilt eggs, but a pathogenic study revealed the effects of AgNPs on hatchlings due to oxidative stress [97]. A recent study done by Kumar et al. [98] showed that AgNPs were successful in reducing the rate of mortality and increase in growth of chicken in two farms in India when applied in poultry drinking water to combat *Escherichia coli* infections. The nanoparticles were of average size, 15 nm, with a minimum inhibitory concentration of 50 mg/L. The research also proved that the poultry used in this research was safe for human consumption, with a hazard quotient of 0.34, which falls in the non-toxic range.

In terms of the use of AgNPs in livestock and poultry production, more in vivo and large-scale research is required, as the existing studies are in vitro, but results are promising in the use of AgNPs as an alternative to antibiotic since the ban of many antibiotics in livestock and poultry industries.

## 5. Toxicity of AgNPs

Toxicity is the root mode of action of AgNPs against pathogens. Toxicity of AgNPs, which is size- and shape-dependent, can inhibit the vital cellular activity of the pathogens at lower concentrations of AgNPs, thus lowering their effects on target organisms. At high concentrations, AgNPs can cause cellular death by interrupting vital cellular activity, causing changes that lead to cellular damage and eventually cellular death [99]. This ability of AgNPs makes them an effective tool against various pathogens and broadens their application in aquaculture and livestock production.

The advantage of the AgNPs toxicity mechanism is their ability to penetrate target organisms with ease due to their large surface-to-volume ratio. This allows them to be excellent antibacterial, antifungal and antiviral agents as they can travel through cellular membranes with ease and increase their bioavailability, thus being able to employ the ‘Trojan Horse’ mechanism [100]. The AgNPs toxicity mechanism also allows for applications as a drug delivery tool where AgNPs can serve as a carrier to ensure that the drugs reach target cells and are released on-site [101].

Although AgNPs offer great benefits, their effects on animal and human health are still in question. The primary disadvantage is the bioaccumulation of AgNPs in aquatic and livestock products. This, in the long run, causes organ damage in animals, and it results in possible human organ damage when consumed, owing to their small size that easily passes through the blood–organ barrier [1]. Another issue is that AgNPs are difficult to remove using common methods of rinsing. They can accumulate in the environment, which may cause changes to the size, shape, surface area and crystalline structure due to environmental factors such as temperature, pH and light intensity. This results in AgNPs of different sizes and shapes with different toxic effects [1,102].

Research into AgNPs effects needs to be conducted further, and in vitro and in vivo applications need to done to understand the extent of negative effects posed by AgNPs to better improve the application process. At the moment, it can be said that the toxicity of AgNPs is unpredictable despite their excellent contribution in the field of nanotechnology.

## 6. Conclusions

The mechanistic action of AgNPs offers an effective and efficient action against pathogens as compared to commercial methods. The application of nanotechnology, primarily AgNPs, in aquaculture and livestock sectors has been proven to solve problems faced by the respective sectors. AgNPs offer a long-term and effective solution in disease control, which in our opinion is the root of the majority of issues faced by the respective industries. An effective solution leads to an increase in aquaculture and livestock production with minimal to no side effects, ensuring the quality of products and, thus, increasing the economic growth of the country as well as the global economy.

## Figures and Tables

**Figure 1 animals-11-02097-f001:**
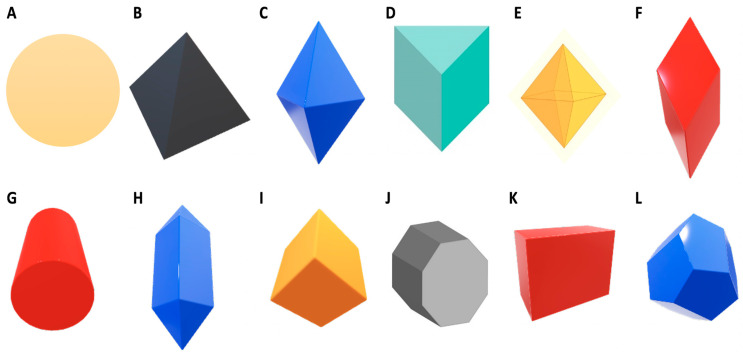
The various shapes of synthesized nanoparticles (adapted from Jo et al. [22] and Walters and Parkin [23]).

**Figure 2 animals-11-02097-f002:**
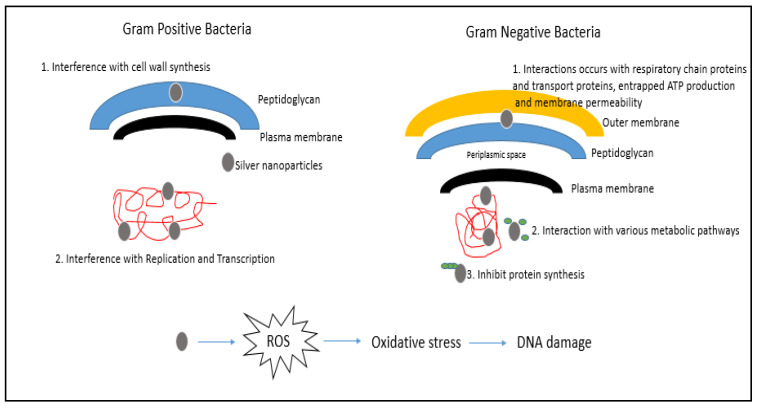
The activation of ROS by AgNPs and the possible mechanism of action towards Gram-negative and Gram-positive bacterial cells (adapted from Pandey et al. [49]).

**Figure 3 animals-11-02097-f003:**
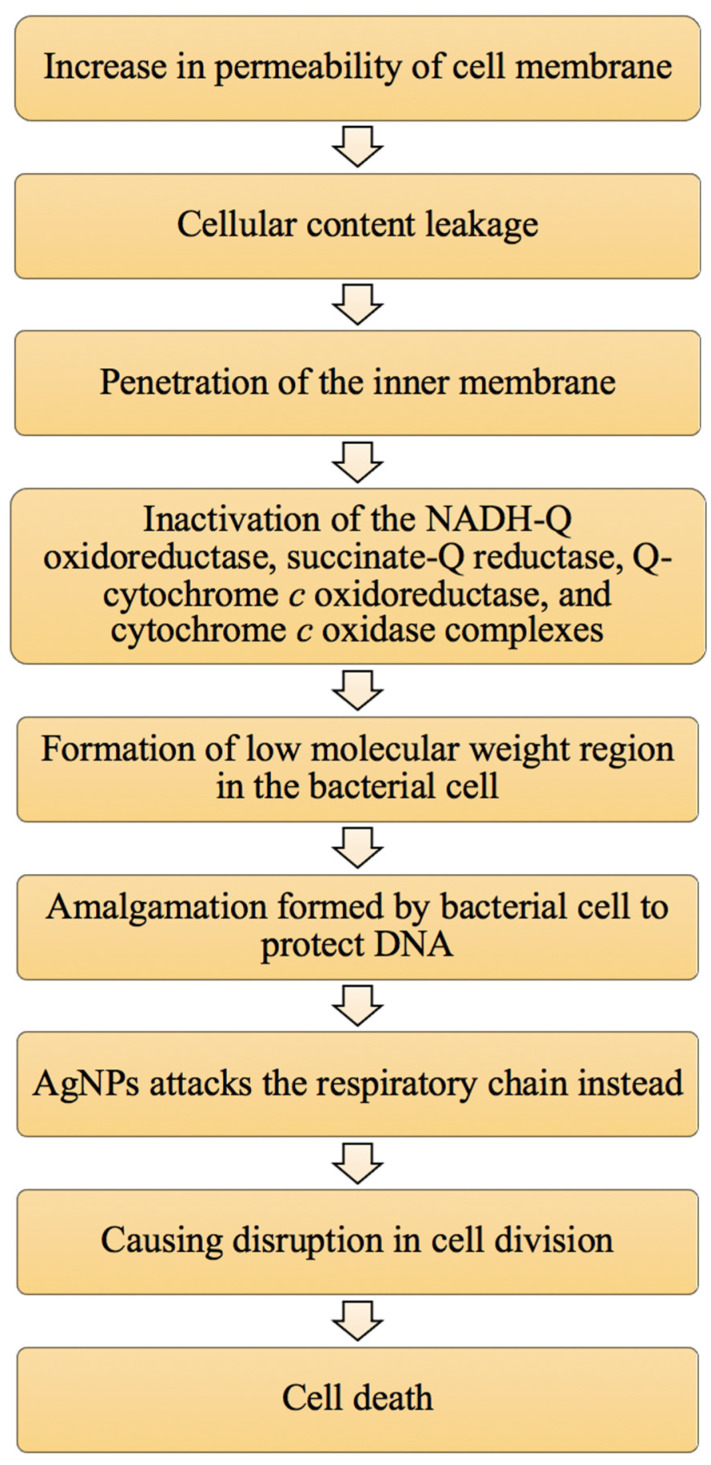
The summary of the pathway of cell death caused by an increase in membrane permeability due to the presence of AgNPs (adapted from Rai et al. [27]).

**Figure 4 animals-11-02097-f004:**
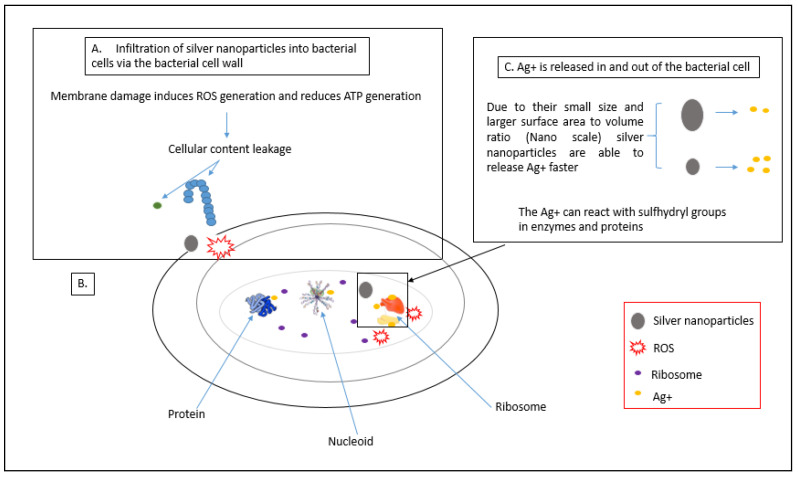
Intracellular damage of the cell caused by AgNPs. (**A**) Process of AgNPs penetration into bacterial cell and disruptions caused on the membrane; (**B**) process of ribosome denaturation by AgNPs; (**C**) interaction of AgNPs with proteins and enzymes, adapted from Qing et al. [63].

**Figure 5 animals-11-02097-f005:**
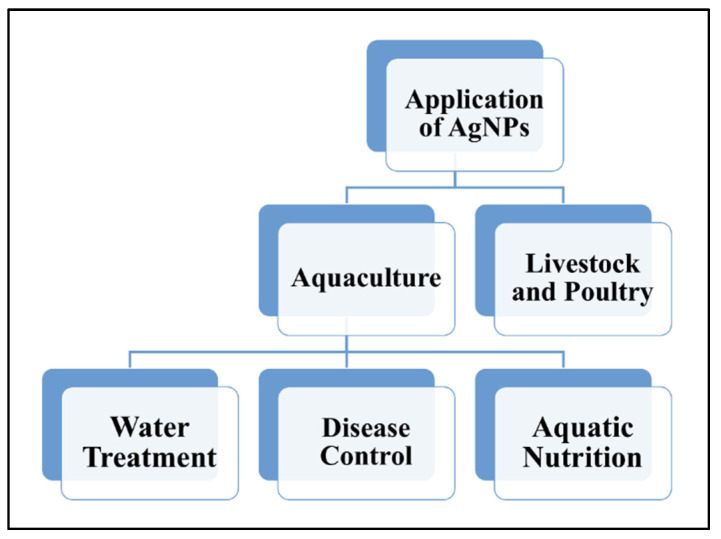
The applications of AgNPs in aquaculture (water treatment, disease control and aquatic nutrients), livestock and poultry industries.

**Figure 6 animals-11-02097-f006:**
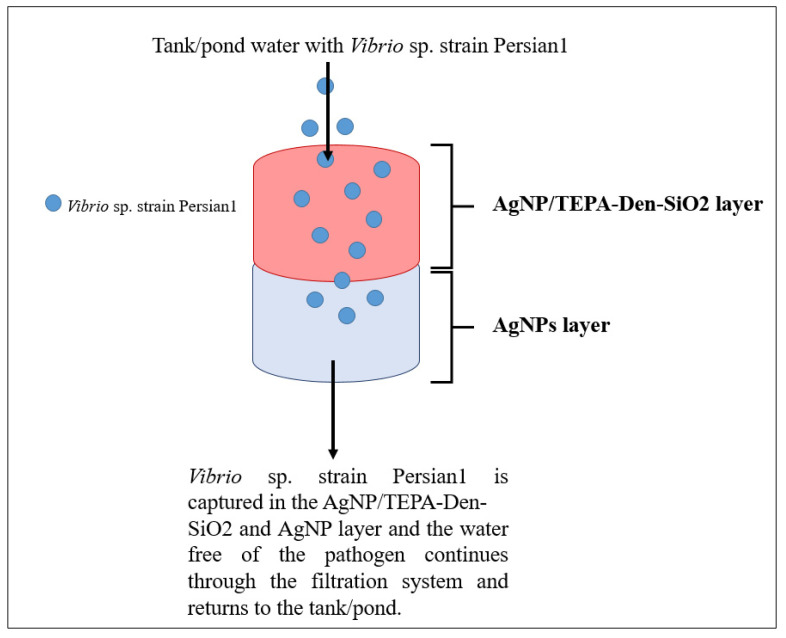
Water filtration system employing AgNPs with a silver absorbent layer (adapted from Sarkheil et al. [76]).

**Figure 7 animals-11-02097-f007:**
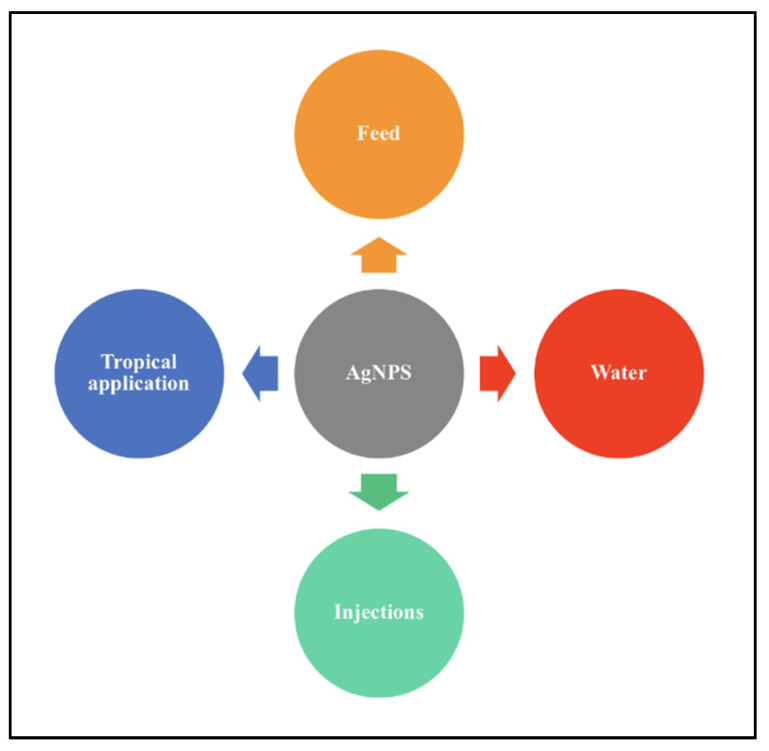
Methods of application of AgNPs on target organisms.

**Table 1 animals-11-02097-t001:** Summary of antibacterial testing done on various strains of pathogenic bacteria and the size of the AgNPs found to be most effective in killing the bacterial strains based on literature.

Bacterial Strain	AgNP Size	Author
*Escherichia coli*	7 nm	[13]
Oral pathogenic bacteria: *Aggregatibacter actinomycetemcomitans*, *Fusobacterium nucleatum*, *Streptococcus mitis*, *Streptococcus mutans*, *Streptococcus sanguinis*. Aerobic: *Escherichia coli*	5 nm	[14]
*Escherichia coli*	55 nm	[15]
Foodborne pathogenic bacteria: *Bacillus cereus* ATCC 13061, *Listeria monocytogenes* ATCC 19115, *Staphylococcus aureus* ATCC 49444, *Escherichia coli* ATCC 43890, and *Salmonella Typhimurium* ATCC 43174 Candida species: *C. albicans* KACC 30003 and KACC 30062, *C. glabrata* KBNO6P00368, *C. geochares* KACC 30061, and *C. saitoana* KACC 41238	31.18 nm, 35.74 nm and 69.14 nm	[16]
Gram-positive bacteria: *Streptococcus* sp., *Bacillus* sp., *Staphylococcus* sp. Gram-negative bacteria: *Shigella* sp., *Escherichia coli, Pseudomonas aeruginosa* and *Klebsiella* sp. Fungus: *Candida* sp.	8.8 nm to 21.4 nm	[17]
*Pseudomonas fluorescens* MTCC 1749, *Proteus mirabilis* MTCC 425, *Escherichia coli* MTCC 1610, *Bacillus cereus and Staphylococcus aureus* MTCC 2940	18 nm to 100 nm and 49 nm to 153 nm	[18]
Gram-negative :*Escherichia coli O157:H7* Gram-positive: *Listeria monocytogenes*	8 nm to 15 nm	[19]

## Data Availability

Not applicable.

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
