# Peer review of "The Mechanistic Action of Biosynthesised Silver Nanoparticles and Its Application in Aquaculture and Livestock Industries"

_animals, 2021, doi:10.3390/ani11072097_

Round 1

Reviewer 1 Report

The authors have provided a review of the latest novel (e.g., aquatic nutrition) information in the literature regarding the use of Ag nanoparticles as antibacterial agents in the livestock and aquaculture industries. Ag NPs can increase survival of aquatic life and can improve immunity in livestock through reduced antibiotic use and production in poultry.

The review covers a wide range of issues with the use of Ag NPs and is well written overall, although it could use a little bit of English idiomatic editing. E.g., "over the current years" in the Simple Summary could be deleted, and "recently" substituted. And use "contributes" vs. "attributes".

Introduction, 1st paragraph: why say "stop here and continue new sentence"? Please delete.

Table 1 is good.

Use of British English (theorized) is fine.

Figures are appropriate, but could even include more detail (Fig. 3--e.g., particular enzymes modulated in the respiratory chain).

Author Response

Comment 1: The review covers a wide range of issues with the use of Ag NPs and is well written overall, although it could use a little bit of English idiomatic editing. E.g., "over the current years" in the Simple Summary could be deleted, and "recently" substituted. And use "contributes" vs. "attributes". Answer: Corrections have been made as mentioned. Line 27 & 31.

Comment 2: Introduction, 1st paragraph: why say "stop here and continue new sentence"? Please delete. Answer: Deleted. Page 2.

Comment 3: Figures are appropriate, but could even include more detail (Fig. 3--e.g., particular enzymes modulated in the respiratory chain). Answer: Corrections have been made as mentioned. Figure 3. Page 7.

Reviewer 2 Report

This review briefly describes the green synthesized AgNPs size and shaped dependent activity, and its extended applications in the aquaculture and livestock. This review was written well and systematically.    

Suggestions

Line no.51. Author should consider writing a few line introductions of metallic nanoparticles before starting the synthesis methods

Line no. 52 nanoparticles should be metallic nanoparticles

Line no. 158 to 167: These are not relevant to the mechanism of action. This should move to size and shape section

Line no.177 is not giving the required meaning. Should be changed

Line no. 233 silver nanoparticles should be AgNPs. Maintain this short form throughout the manuscript

Before conclusion, the author should include the section of toxicity of Ag NPs and include the pros and cons of Ag NPs on the toxicity aspect.

Conclusion should be briefer. Please include the future scope as well as the author’s opinions.  

Author Response

Comment 1: Line no.51. Author should consider writing a few line introductions of metallic nanoparticles before starting the synthesis methods. Answer: Information has been added as mentioned. Page 2: Line 53-57.

Comment 2: Line no. 52 nanoparticles should be metallic nanoparticles. Answer: Corrections have been made as mentioned. Page 2: Line 54.

Comment 3: Line no. 158 to 167: These are not relevant to the mechanism of action. This should move to size and shape section. Answer: The mechanism of action here is centered on the size and shape of the nanoparticle and that is primarily why this information was added to this section. Page 5.

Comment 4: Line no.177 is not giving the required meaning. Should be changed. Answer: Corrections have been made as mentioned. Page 5: Line 179.

Comment 5: Line no. 233 silver nanoparticles should be AgNPs. Maintain this short form throughout the manuscript. Answer: Corrections have been made as mentioned.

Comment 6: Before conclusion, the author should include the section of toxicity of Ag NPs and include the pros and cons of Ag NPs on the toxicity aspect. Answer: A section on the toxicity of AgNPs and the pros and cons have been added. Page 14: Line 483-511.

Comment 7: Conclusion should be briefer. Please include the future scope as well as the author’s opinions.  Answer: Corrections have been made as mentioned. Page 14: Line 516-520.
